# 3DiffTection: 3D Object Detection with Geometry-Aware Diffusion Features

## Abstract

We present 3DiffTection, a state-of-the-art method for 3D object detection from single images, leveraging features from a 3D-aware diffusion model. Annotating large-scale image data for 3D detection is resource-intensive and time-consuming. Recently, pretrained large image diffusion models have become prominent as effective feature extractors for 2D perception tasks. However, these features are initially trained on paired text and image data, which are not optimized for 3D tasks, and often exhibit a domain gap when applied to the target data. Our approach bridges these gaps through two specialized tuning strategies: geometric and semantic. For geometric tuning, we fine-tune a diffusion model to perform novel view synthesis conditioned on a single image, by introducing a novel epipolar warp operator. This task meets two essential criteria: the necessity for 3D awareness and reliance solely on posed image data, which are readily available (e.g., from videos) and does not require manual annotation. For semantic refinement, we further train the model on target data with detection supervision. Both tuning phases employ ControlNet to preserve the integrity of the original feature capabilities. In the final step, we harness these enhanced capabilities to conduct a test-time prediction ensemble across multiple virtual viewpoints. Through our methodology, we obtain 3D-aware features that are tailored for 3D detection and excel in identifying cross-view point correspondences. Consequently, our model emerges as a powerful 3D detector, substantially surpassing previous benchmarks, *e.g.,* Cube-RCNN, a precedent in single-view 3D detection by 9.43% in AP3D on the Omni3D-ARkitscene dataset. Furthermore, 3DiffTection showcases robust data efficiency and generalization to cross-domain data.

## 1 Introduction

Detecting objects in 3D from a single view has long fascinated the computer vision community due to its paramount significance in fields such as robotics and augmented reality. This task requires the computational model to predict the semantic class and oriented 3D bounding box for each object in the scene from a single image with known camera parameters. The inherent challenge goes beyond object recognition and localization to include depth and orientation prediction, demanding significant 3D reasoning capabilities.

Relying on annotated data to train a 3D detector from scratch is not scalable due to the high cost and effort required for labeling (Brazil et al., 2023). Recently, large self-supervised models have emerged as a compelling alternative for image representation learning (He et al., 2021; Chen et al., 2020; He et al., 2020). They acquire robust semantic features that can be fine-tuned on smaller, annotated datasets. Image diffusion models, trained on internet-scale data, have proven particularly effective in this context (Xu et al., 2023b; Li et al., 2023; Tang et al., 2023). However, the direct application of these 2D-oriented advancements to 3D tasks faces inherent limitations. Specifically, these models often lack the 3D awareness necessary for our target task and exhibit domain gaps when applied to the target data.

In an attempt to bridge the 3D gap, recent works have proposed lifting 2D image features to 3D and refining them for specific 3D tasks. NeRF-Det (Xu et al., 2023a) trained a view synthesis model alongside a detection head using pretrained image feature extractors. However, their approach has limited applicability due to the requirement for dense scene views, and the joint training process

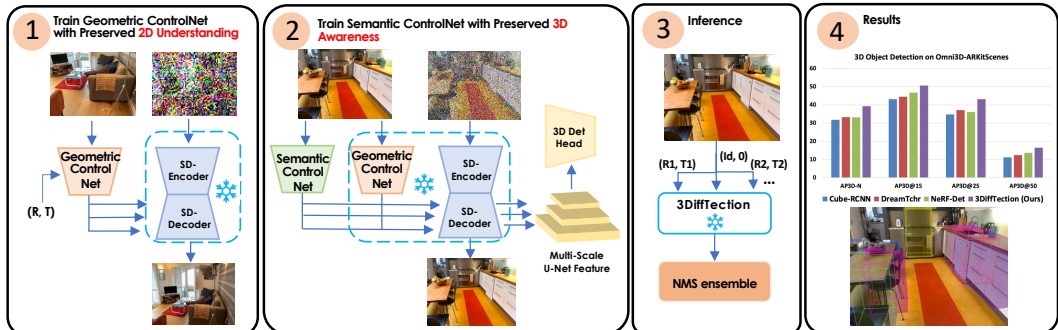

Figure 1: (1) We enhance pre-trained diffusion features with 3D awareness by training a *geometric* ControlNet. (2) We employ a *semantic* ControlNet to refine generative features for specific downstream tasks, with a particular focus on enhancing features for the 3D object detection task. (3) During the inference process, we further enhance 3D detection accuracy by ensembling the bounding boxes from generated views.

necessitates that the data used for reconstruction is fully annotated with detection boxes. Previous efforts in novel view synthesis using diffusion models have shown promise (Chan et al., 2023; Zhou & Tulsiani, 2023). Yet, these models are generally trained from scratch, thereby foregoing the advantages of using pretrained semantic features. To the best of our knowledge, no efforts have been made to leverage these diffusion features in 3D perception tasks.

In this work, we introduce 3DiffTection, a novel framework that enables the use of pretrained 2D diffusion models in 3D object detection tasks (see overview Fig. 1). Key to our approach is a view synthesis task designed to enhance 2D diffusion features with 3D awareness. We achieve this by extracting residual features from source images and warping them to a target view using epipolar geometry. These warped features facilitate the generation of the target output through a denoising diffusion process. Our method capitalizes on image pairs with known relative poses, which are often readily available from video data. Given the ever-increasing availability of video data, this makes our representation refinement solution highly scalable. To demonstrate that this approach successfully endows the model with 3D awareness, we evaluate its features on point correspondence across multiple views and show that it outperforms the base model features.

We proceed to utilize the 3D-enhanced features for 3D detection by training a standard detection head under 3D box supervision. While the baseline performance of our model already shows improvement over existing methods, we aim to further adapt our trained features to the target task and dataset, which may differ from the data used for view synthesis pre-training. Since the training data is limited, attempting to bridge the task and domain gaps by directly fine-tuning the model may result in performance degradation. To address this, we introduce a secondary ControlNet, which helps maintain feature quality (Zhang et al., 2023). This procedure also preserves the model's view synthesis capability. At test time, we capitalize on both geometric and semantic capabilities by generating detection proposals from multiple synthesized views, which are then consolidated through Non-Maximum Suppression (NMS).

Our primary contributions are as follows: (1) We introduce a scalable technique for enhancing pretrained 2D diffusion models with 3D awareness through view synthesis; (2) We adapt these features for a 3D detection task and target domain; and (3) We leverage the view synthesis capability to further improve detection performance through ensemble prediction.

Qualitatively, we demonstrate that our learned features yield improved capabilities in correspondence finding. Quantitatively, we achieve significantly improved 3D detection performance against strong baselines on Omni3D-ARKitscene. Additionally, we illustrate the model's label efficiency, and cross-dataset generalization ability with detection fine-tuning.

## 2 RELATED WORK

### 2.1 3D OBJECT DETECTION FROM IMAGES

3D object detection from posed images is widely explored (Xu et al., 2023a; Rukhovich et al., 2022; Park et al., 2023; Wang et al., 2022b; Liu et al., 2022). However, assuming given camera extrinsic is not a common scenario, especially in applications such as AR/VR and mobile devices. The task of 3D detection from single images, relying solely on camera intrinsics, presents a more generalized

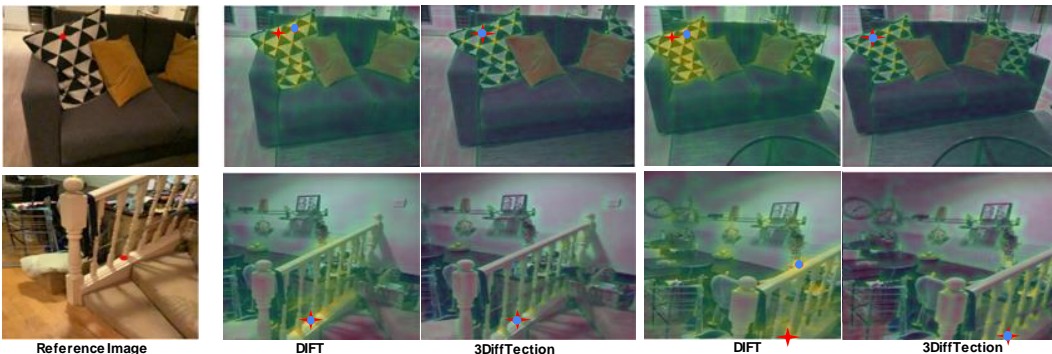


Reference Image      DIFT      3DiffTection      DIFT      3DiffTection


Figure 2: **Visualization of semantic correspondence prediction using different features** Given a **Red Source Point** in the left most reference image, we predict the corresponding points in the images from different camera views on the right (**Blue Dot**). The ground truth points are marked by **Red Stars**. Our method, 3DiffTection, is able to identify precise correspondences in challenging scenes with repetitive visual patterns.

yet significantly more challenging problem. The model is required to inherently learn 3D structures and harness semantic knowledge. While representative methods (Nie et al., 2020; Chen et al., 2021; Huang et al., 2018; Tulsiani et al., 2018; Kulkarni et al., 2019; Wang et al., 2022a) endeavor to enforce 3D detectors to learn 3D cues from diverse geometric constraints, the dearth of semantics stemming from the limited availability of 3D datasets still impede the generalizability of 3D detectors. Brazil et al. (Brazil et al., 2023), in an effort to address this issue, embarked on enhancing the dataset landscape by introducing Omni3D dataset. Rather than focusing on advancing generalizable 3D detection by increasing annotated 3D data, we propose a new paradigm, of enhancing semantic-aware diffusion features with 3D awareness.

## 2.2 DIFFUSION MODELS FOR 2D PERCEPTION

Trained diffusion models (Nichol et al., 2022; Rombach et al., 2022; Ramesh et al., 2022b; Saharia et al., 2022b) have been shown to have internal representations suitable for dense perception tasks, particularly in the realm of image segmentation (Brempong et al., 2022; Xu et al., 2023b; Graikos et al., 2022; Tan et al., 2023). These models demonstrate impressive label efficiency (Baranchuk et al., 2022). Similarly, we observe strong base performance in both 2D and 3D detection (see Tab. 2); our method also benefits from high label efficiency. Diffusion models have further been trained to perform 2D segmentation tasks (Kim et al., 2023; Wolleb et al., 2022; Chen et al., 2022). In (Amit et al., 2021) the model is trained to output a segmentation map using an auxiliary network that outputs residual features. Similarly, we use a ControlNet to refine the diffusion model features to endow them with 3D awareness. We note that several works utilize multiple generations to achieve a more robust prediction (Amit et al., 2021), we go a step further by using our controllable view generation to ensemble predictions from multiple views. Few works have studied tasks other than segmentation. DreamTeacher (Li et al., 2023) proposed to distil the diffusion features to an image backbone and demonstrated excellent performance when tuned on perception tasks(Li et al., 2023). Saxena et al. (2023) trained a diffusion model for dense depth prediction from a single image. Recently, DiffusionDet (Chen et al., 2023) proposed an interesting method for using diffusion models for 2D detection by directly denoising the bounding boxes conditioned on the target image. Diffusion features have been analyzed in (Tumanyan et al., 2023b) showing that different UNet layer activations are correlated with different level of image details. We utilize this property when choosing which UNet layer outputs to warp in our geometric conditioning. Remarkably, (Tang et al., 2023) have shown strong point correspondence ability with good robustness to view change. Here we demonstrate that our 3D-aware features can further boost this robustness.

## 2.3 NOVEL VIEW SYNTHESIS WITH DIFFUSION MODELS

Image synthesis has undergone a significant transformation with the advent of 2D diffusion models, as demonstrated by notable works (Sohl-Dickstein et al., 2015; Ho et al., 2020; Song et al., 2021; Nichol & Dhariwal, 2021; Dhariwal & Nichol, 2021; Ho et al., 2021; Nichol et al., 2021; Rombach et al., 2022; Ramesh et al., 2022a; Saharia et al., 2022a). These models have extended their capabilities to the Novel View Synthesis (NVS) task, where 3DiM (Watson et al., 2022) and Zero-123 (Liu et al., 2023) model NVS of objects as a viewpoint-conditioned image-to-image translation task with diffusion models. The models are trained on a synthetic dataset with camera annotation and

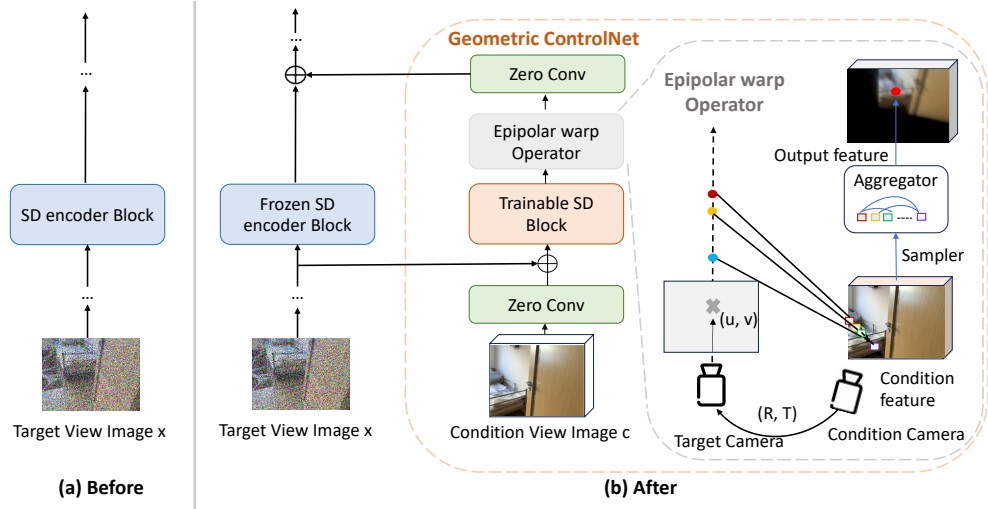

Figure 3: **Architecture of Geometric ControlNet. Left**: Original Stable Diffusion UNet encoder block. **Right**: We train novel view image synthesis by adding a geometric ControlNet to the original Stable Diffusion encoder blocks. The geometric ControlNet receives the conditional view image as an additional input. Using the camera pose, we introduce an epipolar warp operator, which warps intermediate features into the target view. With the geometric ControlNet, we significantly improve the 3D awareness of pre-trained diffusion features.

demonstrate zero-shot generalization to in-the-wild images. NerfDiff (Gu et al., 2023) distills the knowledge of a 3D-aware conditional diffusion model into a Nerf. RealFusion (Melas-Kyriazi et al., 2023) uses a diffusion model as a conditional prior with designed prompts. NeuralLift (Xu et al., 2022) uses language-guided priors to guide the novel view synthesis diffusion model. Most recently, inspired by the idea of video diffusion models (Singer et al., 2022; Ho et al., 2022; Blattmann et al., 2023), MVDream (Shi et al., 2023) adapts the attention layers to model the cross-view 3D dependency. The most relevant work to our approaches is SparseFusion (Zhou & Tulsiani, 2023), where authors propose to incorporate geometry priors with epipolar geometries. However, while their model is trained from scratch, in our approach, we use NVS merely as an auxiliary task to enhance the pre-trained diffusion features with 3D awareness and design the architecture for tuning a minimal number of parameters by leveraging a ControlNet.

## 3 METHOD

### 3.1 OVERVIEW

We introduce 3DiffTection, designed to harness diffusion model features for 3D detection. As depicted in Fig. 1, 3DiffTection comprises three core components: 1) Instilling 3D awareness into the diffusion features by training a geometric ControlNet for view synthesis. 2) Bridging the domain and task gaps using a semantic ControlNet, which is concurrently trained with a 3D detection head on the target data distribution. 3) Amplifying 3D box predictions through a virtual view ensembling strategy. We will further detail each of these steps in the subsequent sections.

### 3.2 DIFFUSION MODEL AS A FEATURE EXTRACTOR

Recent works demonstrate that features extracted from text-to-image diffusion models, such as Stable Diffusion (Rombach et al., 2022), capture rich semantics suitable for dense perception tasks, including image segmentation (Xu et al., 2023b) and point correspondences (Tang et al., 2023). In this work, our interest lies in 3D object detection. However, since Stable Diffusion is trained on 2D image-text pairs—a pre-training paradigm proficient in aligning textual semantics with 2D visual features—it might lack 3D awareness. We aim to explore this by examining point correspondences between views. We hypothesize that features with 3D awareness should demonstrate the capability to identify correspondences that point to the same 3D locations when provided with multi-view images.

Following (Xu et al., 2023b; Tang et al., 2023) we employ a single forward step for feature extraction. However, unlike these works, we only input images without textual captions, given that in real-world scenarios, textual input is typically not provided for object detection. Formally, given an image $\mathbf{x}$, we sample a noise image $\mathbf{x}_t$ at time $t$, and obtain the diffusion features as

$$\mathbf{f} = \mathcal{F}(\mathbf{x}_t; \Theta), \mathbf{x}_t = \sqrt{\bar{\alpha}_t}\mathbf{x} + \sqrt{1 - \bar{\alpha}_t}\epsilon_t, \epsilon_t \sim \mathbb{N}(0, 1), \tag{1}$$

where $\mathbf{f}$ represents the multi-scale features from the decoder module of UNet $\mathcal{F}$ (parameterized by $\Theta$), and $\alpha_t$ represents a pre-defined noise schedule, satisfying $\bar{\alpha}_t = \prod_{k=1}^{t} \alpha_k$.

Interestingly, as illustrated in Fig. 2, the point localization of Stable Diffusion features depends on 2D appearance matching. This can lead to confusion in the presence of repeated visual patterns, indicating a deficiency in 3D spatial understanding. Given this observation, we aim to integrate 3D awareness into the diffusion features, which we will discuss in the following section.

### 3.3 INCORPORATING 3D AWARENESS TO DIFFUSION FEATURES.

**ControlNet** (Zhang et al., 2023) is a powerful tool that allows the addition of conditioning into a pre-trained, static Stable Diffusion (SD) model. It has been demonstrated to support various types of dense input conditioning, such as depth and semantic images. This is achieved through the injection of conditional image features into trainable copies of the original SD blocks. A significant attribute of ControlNet is its ability to resist overfitting to the dataset used for tuning while preserving the original model's performance. As a result, ControlNet is well-suited for enhancing diffusion features with 3D awareness without compromising their 2D semantic quality.

Formally, we denote one block of UNet $\mathcal{F}$ as $\mathcal{F}_s(\cdot; \Theta_s)$ parameterized by $\Theta_s$. In particular, the original ControlNet block copies each pre-trained Stable Diffusion module $\mathcal{F}_s(\cdot; \Theta_s)$ denoted as $\mathcal{F}'_s(\cdot; \Theta'_s)$, and accompanying with two zero convolutions $\mathcal{Z}_{s1}$ and $\mathcal{Z}_{s2}$, parameterized by $\Theta_{zs1}$ and $\Theta_{zs2}$, respectively. We slightly abuse the notation of $\mathbf{x} \in \mathcal{R}^{H \times W \times C}$ as the arbitrary middle features of $\mathbf{x}_t$ in $\mathcal{F}$. Then a ControlNet block with the corresponding frozen Stable Diffusion block is given by

$$\mathbf{y}_s = \mathcal{F}_s(\mathbf{x}; \Theta_s) + \mathcal{Z}_{s2}(\mathcal{F}'_s(\mathbf{x} + \mathcal{Z}_{s1}(\mathbf{c}; \Theta_{zs1}); \Theta'_s); \Theta_{zs2}), \qquad (2)$$

where $\mathbf{c} \in \mathcal{R}^{H \times W \times C}$ is the condition image feature and $\mathbf{y}_s \in \mathcal{R}^{H \times W \times C}$ is the output.

**Epipolar warp operator.** We utilize ControlNet to enhance the 3D awareness of diffusion features by training it to perform view synthesis. Specifically, we select pairs of images with known relative camera poses and train the ControlNet conditioned on the source view to produce the output view. Since the features induced by the condition in ControlNet are additive, it is a common practice to ensure alignment between these features and the noisy input features. However, the input for our view synthesis task is, by definition, not aligned with the noisy input of the target view. As a solution, we propose to warp the source view features to align with the target using epipolar geometry. We denote the epipolar warp operator as $\mathcal{G}(\cdot, T_n)$, and our *geometric* ControlNet is formulated as:

$$\mathbf{y}_s = \mathcal{F}_s(\mathbf{x}; \Theta_s) + \mathcal{Z}_{s2}(\mathcal{G}(\mathcal{F}'_s(\mathbf{x} + \mathcal{Z}_{s1}(\mathbf{c}; \Theta_{zs1}); \Theta'_s), T_n); \Theta_{zs2}), \qquad (3)$$

Formally, to obtain the target novel-view image at position $(u, v)$, we assume that relative camera extrinsic from the source view is described by $T_n = [[R_n, 0]^T, [t_n, 1]^T]$, and the intrinsic parameters are represented as $K$. The equation for the epipolar line is:

$$l_c = K^{-T}([t_n] \times R_n)K^{-1}[u, v, 1]^T, \qquad (4)$$

Here, $l_c$ denotes the epipolar line associated with the source conditional image. We sample a set of features along the epipolar line, denoted as $\{\mathbf{c}(p_i)\}$, where the $p_i$ are points on the epipolar line. These features are then aggregated at the target view position $(u, v)$ via a differentiable aggregator function, resulting in the updated features:

$$\mathbf{c}'(u, v) = \text{aggregator}(\{\mathbf{c}(p_i)\}), \quad p_i \sim l_c. \qquad (5)$$

The differentiable aggregator can be as straightforward as average/max functions or something more complex like a transformer, as demonstrated in (Zhou & Tulsiani, 2023; Du et al., 2023), and $\mathbf{c}'$ is the warped condition image features, *i.e.*, the output of epipolar warp operator $\mathcal{G}$. The geometric warping procedure is illustrated in Fig. 3.

Interestingly, we found it beneficial to avoid warping features across all the UNet decoder blocks. As highlighted by (Tumanyan et al., 2023a), middle-layer features in Stable Diffusion emphasize high-level semantics, while top stages capture appearance and geometry. Given the shared semantic content in novel-view synthesis, even amidst pixel deviations, we warp features only in the final two stages of Stable-Diffusion. This maintains semantic consistency while accommodating geometric warping shifts. Our geometric ControlNet notably enhances the 3D awareness of diffusion features, evident in the 3DiffTection examples in Fig. 2.

### 3.4 BRIDGING THE TASK AND DOMAIN GAP

We leverage the 3D-enhanced features for 3D detection by training a standard detection head with 3D box supervision. To further verify the efficacy of our approach in adapting diffusion features for 3D tasks, we train a 3D detection head, keeping our fine-tuned features fixed. Notably, we observe a substantial improvement compared to the baseline SD feature. We report details in Tab. 2.

Nevertheless, we acknowledge two potential gaps. Firstly, our view synthesis tuning is conceptualized as a universal 3D feature augmentation method. Hence, it is designed to work with a vast collection of posed image pairs, which can be inexpensively gathered (e.g., from videos) without the need for costly labeling. Consequently, there might be a domain discrepancy when comparing to target data, which could originate from a smaller, fully annotated dataset. Secondly, since the features aren't specifically fine-tuned for detection, there is further potential for optimization towards detection, in tandem with the detection head. As before, we aim to retain the robust feature characteristics already achieved and choose to deploy a second ControlNet.

Specifically, we freeze both the original SD and the geometric ControlNet modules. We then introduce another trainable ControlNet, which we refer to as *semantic* ControlNet. For our model to execute single-image 3D detection, we utilize the input image $x$ in three distinct ways. First, we extract features from it using the pretrained SD, denoted as $\mathcal{F}(x)$, through a single SD denoising forward step. Next, we feed it into our geometric ControlNet, represented as $\mathcal{F}_{geo}(x, T_n)$, with an identity pose ($T_n = [Id, 0]$) to obtain our 3D-aware features. Lastly, we introduce it to the semantic ControlNet, denoted by $\mathcal{F}_{sem}(x)$, to produce trainable features fine-tuned for detection within the target data distribution. We aggregate all the features and pass them to a standard 3D detection head, represented as $\mathcal{D}$ (Brazil et al., 2023). The semantic ControlNet is trained with 3D detection supervision.

$$y = \mathcal{D}(\mathcal{F}(x) + \mathcal{F}_{geo}(x, [Id, 0]) + \mathcal{F}_{sem}(x)) \tag{6}$$

The figure overview is Fig. 6 in the supplementary material.

### 3.5 ENSEMBLE PREDICTION

ControlNet is recognized for its ability to retain the capabilities of the pre-tuned model. As a result, our semantically tuned model still possesses view synthesis capabilities. We exploit this characteristic to introduce a test-time prediction ensembling technique that further enhances detection performance.

Specifically, our box prediction $y$ is dependent on the input view. Although our detection model is trained with this pose set to the identity (i.e., no transformation), at test time, we can incorporate other viewing transformations denoted as $\xi_i$,

$$y(\xi) = \mathcal{D}(\mathcal{F}(x) + \mathcal{F}_{geo}(x, \xi) + \mathcal{F}_{sem}(x)). \tag{7}$$

The final prediction is derived through a non-maximum suppression of individual view predictions:

$$y_{final} = NMS(\{y(\xi_i)\}). \tag{8}$$

We note that our objective isn't to create a novel view at this stage but to enrich the prediction using views that are close to the original pose. The underlying intuition is that the detection and view synthesis capabilities complement each other. Certain objects might be localized more precisely when observed from a slightly altered view.

## 4 EXPERIMENTS

In this section, we conduct experimental evaluations of 3DiffTection, comparing it to previous baselines on the Omni3D dataset. We perform ablation studies to analyze various modules and design choices. Then we demonstrate the effectiveness of our model under data scarcity settings.

**Datasets and implementation details** For all our experiments, we train the geometric ControlNet on the official ARKitscene datasets (Baruch et al., 2021), which provide around 450K posed low-resolution ($256 \times 256$) images. We sample around 40K RGB images along with their intrinsics and extrinsics. For training 3D object detection, we use the official Omni3D dataset (Brazil et al., 2023), which is a combination of sampled ARKitScenes, SUN-RGBD, HyperSim and two autonomous driving datasets. We use Omni3D-ARkitScenes as our primary in-domain experiment dataset, and Omni3D-SUN-RGBD and Omni3D-indoor for our cross-dataset experiments. To evaluate the performance, we compute a **mean** AP3D across all categories in Omni3D-ARkitScenes and over a

| Methods | Resolution | NVS Train Views | Det. Train Views | AP3D↑ | AP3D@15↑ | AP3D@25↑ | AP3D@50↑ |
|---------|-----------|-----------------|------------------|-------|----------|----------|----------|
| CubeRCNN-DLA | 256×256 | - | 1 | 31.75 | 43.10 | 34.68 | 11.07 |
| DreamTchr-Res50 | 256×256 | - | 1 | 33.20 | 44.54 | 37.10 | 12.35 |
| NeRF-Det-R50 | 256×256 | > 2 | > 2 | 33.13 | 46.81 | 36.03 | 13.58 |
| ImVoxelNet | 256×256 | - | > 2 | 32.09 | 46.71 | 35.62 | 11.94 |
| 3DiffTection | 256×256 | 2 | 1 | **39.22** | **50.58** | **43.18** | **16.40** |
| CubeRCNN-DLA | 512×512 | - | 1 | 34.32 | 46.06 | 36.02 | 12.51 |
| DreamTchr-Res50 | 512×512 | - | 1 | 36.14 | 49.82 | 40.51 | 15.48 |
| 3DiffTection | 512×512 | 2 | 1 | **43.75** | **57.13** | **47.32** | **20.30** |
| CubeRCNN-DLA-Aug | 512×512 | - | 1 | 41.72 | 53.09 | 45.42 | 19.26 |

Table 1: **3D Object Detection Results on Omni3D-ARKitScenes testing set**. 3DiffTection significantly outperforms baselines, including CubeRCNN-DLA-Aug, which is trained with 6x more supervision data.

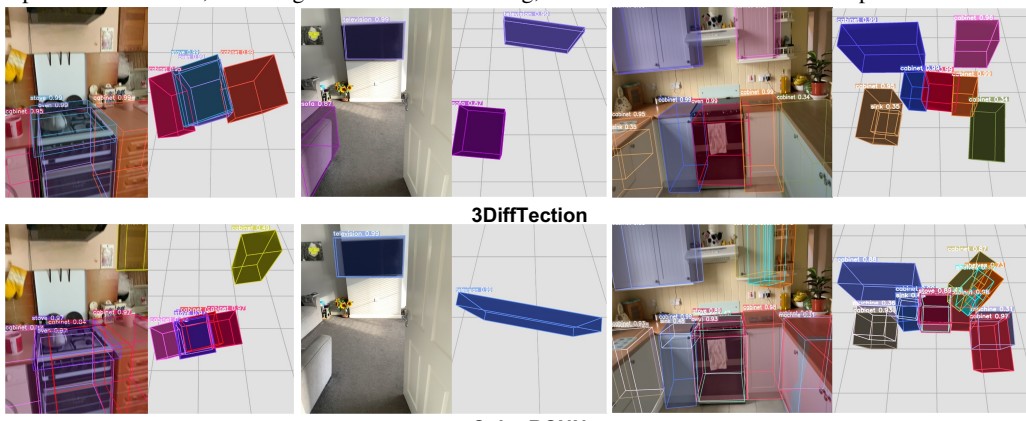

**3DiffTection**

**Cube-RCNN**

Figure 4: **Qualitative results on Omni3D-ARKitScene 3D Detection.** In contrast to Cube-RCNN (bottom), our approach (top) accurately predicts both the box class and 3D locations. The bird's-eye-view visualization further demonstrates that our predictions surpass the baseline performance of Cube-RCNN.

range of IoU3D thresholds in $[0.05, 0.10, ..., 0.50]$, simply denoted as **AP3D**. We also report AP3D at IoU 15, 25, and 50 (AP3D@15, AP3D@25 and AP3D@50) as following (Brazil et al., 2023). We take the publicly available text-to-image LDM (Rombach et al., 2022), Stable Diffusion as the preliminary backbone. Unlike previous diffusion models which require multiple images for training a novel-view synthesis task, we only take *two* views, one as the source view and another one as the target view. Moreover, we only consider two views with an overlap of less than 30%. Regarding novel-view synthesis ensemble, we use pseudo camera rotations, *i.e.*, $\pm 15 \deg$ and ensemble the predicted bounding boxes via NMS.

**Methods in comparison.**    CubeRCNN (Brazil et al., 2023) extends Fast-RCNN (Ren et al., 2015) to 3D object detection by incorporating a cube head. In our work, we aim to provide a stronger 3D-aware image backbone, and compare it with other image backbones using the Cube-RCNN framework. Specifically, we compare with DreamTeacher (Li et al., 2023), which distills knowledge from a Pre-trained Stable Diffusion to a lighter network, *ResNet-50*. We also compare with DIFT (Tang et al., 2023), which directly employs the frozen Stable Diffusion as the image feature extractor. Additionally, we evaluate methods designed for multi-view 3D detection, such as NeRF-Det (Xu et al., 2023a) and ImVoxelNet (Rukhovich et al., 2022). While these methods typically require more images during training, we use them for single-image 3D object detection during testing.

### 4.1 3D Object Detection on Omni3D-ARKitScenes

In Table 1, we analyze the 3D object detection performance of 3DiffTection in comparison to several baseline methods. Notably, 3DiffTection outperforms CubeRCNN-DLA (Brazil et al., 2023), a prior art in single-view 3D detection on the Omni3D-ARKitScenes dataset, by a substantial margin of 7.4% at a resolution of 256 × 256 and 9.43% at a resolution of 512 × 512 on the AP3D metric. We further compare our approach to NeRF-Det-R50 (Xu et al., 2023a) and ImVoxelNet (Rukhovich et al., 2022), both of which utilize multi-view images during training (indicated in Tab. 1 as NVS Train Views and Det. Train Views). In contrast, 3DiffTection does not rely on multi-view images for training the detection network and uses only 2 views for the geometric network training, outperforming the other methods by 6.09% and 7.13% on the AP3D metric, respectively. Additionally, we compare our approach to DreamTeacher-Res50 (Li et al., 2023), which distills StableDiffusion feature prediction into a ResNet backbone to make it amenable for perception tasks. 3DiffTection

| Backbone | NVS Train Views | Geo-Ctr | Sem-Ctr | NV-Ensemble | AP2D | AP3D↑ | AP3D@15↑ | AP3D@25↑ | AP3D@50↑ |
|---|---|---|---|---|---|---|---|---|---|
| VIT-B (MAE) | - | - | - | - | 26.14 | 25.23 | 36.04 | 28.64 | 8.11 |
| Res50 (DreamTchr) | - | - | - | - | 25.27 | 24.36 | 34.16 | 25.97 | 7.93 |
| StableDiff. (DIFT) | - | - | - | - | 29.35 | 28.86 | 40.18 | 32.07 | 8.86 |
| StableDiff. (Ours) | 1 | ✓ | - | - | 29.51 | 26.05 | 35.81 | 29.86 | 6.95 |
| StableDiff. (Ours) | 2 | ✓ | - | - | 30.16 | 31.20 | 41.87 | 33.53 | 10.14 |
| StableDiff. (Ours) | 2 | ✓ | ✓ | - | 37.12 | 38.72 | 50.38 | 42.88 | 16.18 |
| StableDiff. (Ours) | 2 | ✓ | ✓ | ✓ | 37.19 | 39.22 | 50.58 | 43.18 | 16.40 |

Table 2: **Analysis of 3DiffTection Modules on Omni3D-ARKitScenes testing set**. We first compare different backbones by freezing the backbone and only training the 3D detection head. Then, we perform ablative studies on each module of our architecture systematically. Starting with the baseline vanilla stable diffusion model, we incrementally incorporate improvements: Geometry-ControlNet (**Geo-Ctr**), the number of novel view synthesis training views (**NVS Train Views**), Semantic-ControlNet (**Sem-Ctr**), and the novel view synthesis ensemble (**NV-Ensemble**).

| Methods | Backbone | Pre-training Data | Pre-training Task | Trained Module | SUNRGBD | Omni-indoor |
|---|---|---|---|---|---|---|
| DIFT-SD | StableDiff | LION5B (Schuhmann et al., 2022) | Generation | 3D Head | 15.35 | 15.94 |
| CubeRCNN | DLA34 | ImageNet & Aktscn | Classification & 3D det. | 3D Head | 14.68 | 16.13 |
| 3DiffTection | StableDiff+Geo-Ctr | LION5B & Aktscn | Generation | 3D Head | 16.68 | 17.21 |
| 3DiffTection | StableDiff+Geo-Ctr | LION5B & Aktscn | Generation | Sem-Ctr+3D Head | 19.01 | 22.71 |

Table 3: **Cross-Domain experiment on Omni3D-SUNRGBD and Omni3D-indoor dataset 3D detection.** We train 3DiffTection's geometric ControlNet on Omni3D-ARKitScenes (Aktscn) training set and test on Omni3D-SUNRGBD and Omni3D-Indoor dataset. 3DiffTection outperforms baselines with only 3D head training. The results are reported based on AP3D@15.

surpasses DreamTeacher by 6.02% and 7.61% at resolutions of $256 \times 256$ and $512 \times 512$, respectively. Finally, we assess our model against CubeRCNN-DLA-Aug, which represents CubeRCNN trained on the complete Omni3D dataset, comprising 234,000 RGB images with a stronger training recipe. Notably, our model outperforms CubeRCNN-DLA-Aug by 2.03% on AP3D while using nearly 6x less data, demonstrating its data efficiency and effectiveness.

## 4.2 ANALYSIS AND ABLATION

**3DiffTection modules** We analyze the unique modules and design choices in 3DiffTection: the Stable Diffusion backbone, geometric and semantic ControlNets targeting NVS and detection, and the multi-view prediction ensemble. All results are reported using the Omni3D-ARKitScenes in Table 2. We first validate our choice of using a Stable Diffusion backbone. While diffusion features excel in 2D segmentation tasks (Li et al., 2023; Xu et al., 2023b), they have not been tested in 3D detection. We analyze this choice independently from the other improvements by keeping the backbone frozen and only training the 3D detection head. We refer to this setup as *read-out*. The vanilla Stable Diffusion features yield a 28.86% AP3D, surpassing both the CubeRCNN-VIT-B backbone (with MAE pretraining (He et al., 2021)) by 3.63% and the ResNet-50 DreamTeacher backbone by 4.5% on AP30. Similar trends are observed in the AP2D results, further confirming that the Stable Diffusion features are suitable for perception tasks. Our geometric ControlNet, is aimed at instilling 3D awareness via NVS training. We evaluate its performance under the read-out setting and show it boosts performance by 2.34% on AP3D and 0.81% on AP2D. This indicates that the geometric ControlNet imparts 3D awareness knowledge while preserving its 2D knowledge. To ensure our improvement is attributed to our view synthesis training, we limited the geometric ControlNet to single-view data by setting the source and target views to be identical (denoted by '1' in the NVS train view column of Tab. 2), which reduces the training to be *denoising training* (Brempong et al., 2022). The findings indicate a 2.81% decrease in AP3D compared to the standard Stable Diffusion, affirming our hypothesis. Further, the semantic ControlNet, co-trained with the 3D detection head enhances both AP2D and AP3D by roughly 7% confirming its efficacy in adapting the feature for optimal use by the detection head. Lastly, using NVS-ensemble results in an additional 0.5% increase in AP3D demonstrating its role in improving 3D localization.

**Cross-dataset experiments** To assess the capability of 3DiffTection's geometric ControlNet to carry its 3D awareness to other datasets, we employed a 3DiffTection model with its geometric ControlNet trained on the ARKitscene dataset and trained only a 3D head on cross-domain datasets. As a baseline, we trained the 3D head using DIFT-SD features. The results are shown in Tab. 7. In this setup, 3DiffTection outperformed DIFT-SD by 1.33% and 1.27% respectively. We further compared our approach with CubeRCNN. To ensure a fair comparison, we took CubeRCNN-DLA trained on Omni3D-ARKitscene datasets and further fine-tuned its entire model on the Omni3D-

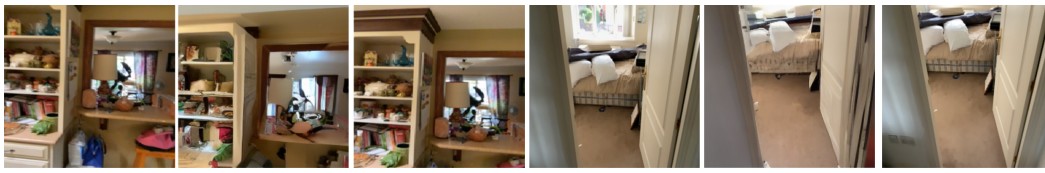

Condition Image    Generated Image    GT      Condition Image    Generated Image    GT

Figure 5: **Novel-view synthesis visualization on Omni3D-ARKitScenes testing set.** Our model with Geometry-ControlNet synthesizes realistic novel views from a single input image.

SUNRGBD and Omni-indoor datasets. Without any training of the geometric ControlNet on the target domain, 3DiffTection surpassed the fully fine-tuned CubeRCNN-DLA by 2.0% and 1.08%. Finally, we reintegrated the semantic ControlNet and jointly trained it with the 3D head. This yielded a performance boost of 2.33% and 5.5%. These results indicate that even without training the geometric ControlNet in the target domain, the semantic ControlNet adeptly adapts features for perception tasks.

**Label efficiency** We hypothesize that our usage of semantic ControlNet for tuning 3DiffTection towards a target dataset should maintain high label efficiency. We test this by using 50% and 10% labels from the Omni3D-ARKitscene datasets. The results are shown in Tab. 4 of supplementary materials. In low-data regime (for both 50% and 10% label setting), 3DiffTection demonstrates significantly better performance, and more modest degradation than baselines. Notably, even with 50% of the labels, our proposed 3DiffTection achieves 2.28 AP3D-N improvement over previous methods trained on 100% label. Additionally, when tuning only the 3D head 3DiffTection performs better than CubeRCNN and DreamTeacher with tuning all parameters.

### 4.3 QUALITATIVE RESULTS AND VISUALIZATIONS

**3D Detection visualization (Fig. 4)** Compared to CubeRCNN, our proposed 3DiffTection predicts 3D bounding boxes with better pose, localization and significantly fewer false defections. As seen in the middle column, our model can even handle severe occlusion cases, *i.e.,* the sofa.

**Feature correspondence visualization (Fig. 2)** As described in 3.2, we conducted a feature correspondence experiment. As can be seen, our method yields a more accurate point-matching result, primarily because our geometric ControlNet is trained to infer 3D correspondences through our Epipolar warp operator to successfully generate novel views. To provide further insights, we visualize a heatmap demonstrating the similarity of target image features to the reference key points. Notably, our 3DiffTection features exhibit better concentration around the target point.

**Novel-view synthesis visualization (Fig. 5)** To validate our geometric ControlNet ability to maintain geometric consistency of the source view content, we visualize novel-view synthesis results. The results demonstrate that our proposed epipolar warp operator is effective in synthesizing the scene with accurate geometry and layout compared to the ground truth images. We note that scene-level NVS from a single image is a challenging task, and we observe that our model may introduce artifacts. While enhancing performance is an interesting future work, here we utilize NVS as an auxiliary task which is demonstrated to effectively enhance our model's 3D awareness.

## 5 CONCLUSION AND LIMITATIONS

We introduced 3DiffTection for 3D detection from a single image that leverages features from a 3D-aware diffusion model. This method effectively addresses the challenges of annotating large-scale image data for 3D object detection. By incorporating geometric and semantic tuning strategies, we have enhanced the capabilities of existing diffusion models, ensuring their applicability to 3D tasks. Notably, our method significantly outperforms previous benchmarks and exhibits high label efficiency and strong adaptability to cross-domain data.

**Limitations.** We highlight several limitations in 3DiffTection. First, even though our geometric feature tuning does not require box annotations, it does necessitate image pairs with accurate camera poses. Extracting such poses from in-the-wild videos can introduce errors which may require additional handling by our method. Furthermore, in-the-wild footage often contains dynamic objects, which our current method does not address. Lastly, by employing the Stable Diffusion architecture, we introduce a significant demand on memory and runtime. Our method achieves a speed of approximately 7.5 fps on a 3090Ti GPU (see details in the Appendix). While 3DiffTection is apt for

offline object detection tasks (e.g., auto-labeling), further refinement is needed to adapt it for online detection settings.

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

## A  APPENDIX

### A.1  MORE IMPLEMENTATION DETAILS.

The Omni3D-ARKitScenes include 51K training images, 7.6K test images and total 420K oriented 3D bounding boxes annotation. Omni3D-SUN-RGBD includes 5.2K training images, 5K test images and total 40K 3D bounding box annotations. Omni-Indoor is a combination of ARKitScenes, SUN-RGBD and HyperSim datasets. Our geometric ControlNet is trained at a resolution of $256 \times 256$, following the camera pose annotations provided by the ARKitScenes dataset (Baruch et al., 2021). The novel-view synthesis training is mainly developed based on ControlNet implementation from Diffusers [1]. We use the same training recipe as Diffusers. To report results at a resolution of $512 \times 512$, we directly input images with a resolution of $512 \times 512$ into the backbone. Note that different to both original ControlNet (Zhang et al., 2023) and DIFT (Tang et al., 2023) use text as input, we input empty text into the diffusion model thorough the experiments.

### A.2  3D-DETECTION HEAD AND OBJECTIVE.

**3D detection head.**  We use the same 3D detection head as Cube-RCNN (Brazil et al., 2023). Specifically, Cube-RCNN extends Faster R-CNN with a cube head to predict 3D cuboids for detected 2D objects. The cube head predicts category-specific 3D estimations, represented by 13 parameters including (1) projected 3D center (u, v) on the image plane relative to the 2D Region of Interest (RoI); (2) object's center depth in meters, transformed from virtual depth (z); (3) log-normalized physical box dimensions in meters $(\bar{w}, \bar{h}, \bar{l})$; (4) Continuous 6D allocentric rotation $(p \in \mathcal{R})$; and (5) predicted 3D uncertainty $\mu$. With parameterized by the output of the cube head, the 3D bounding boxes can be represented by

$$B_{3D}(u, v, z, \bar{w}, \bar{h}, \bar{l}, p) = R(p) \cdot d(\bar{w}, \bar{h}, \bar{l}) \cdot B_{\text{unit}} + X(u, v, z), \qquad (9)$$

where R(p) is the rotation matrix and $d$ is the 3D box dimensions, parameterized by $\bar{w}, \bar{h}, \bar{l}$. $X(u, v, z)$ is the bounding box center, represented by

$$X(u, v, z) = \left(\frac{z}{f_x}\right)\left(rx + u \cdot r_w - p_x, \frac{z}{f_y}\right)(ry + v \cdot r_h - p_y), \qquad (10)$$

where: $[r_x, r_y, r_w, r_h]$ are the object's 2D bounding box. $(f_x, f_y)$ are the known focal lengths of the camera. $(p_x, p_y)$ represents the principal point of the camera. Given the representation of the 3D bounding box, our detection training objective is

$$L = L_{\text{RPN}} + L_{2D} + \sqrt{2} \cdot \exp(-\mu) \cdot L_{3D} + \mu, \qquad (11)$$

where $L_{\text{RPN}}$ and $L_{2D}$ are commonly used in 2D object detection such as Faster-RCNN (Ren et al., 2015), here $L_{3D}$ is given by

$$L(u, v)_{3D} = \|B_{3D}(u, v, z_{\text{gt}}, \bar{w}_{\text{gt}}, \bar{h}_{\text{gt}}, \bar{l}_{\text{gt}}, p_{\text{gt}}) - B_{\text{gt}}^{3D}\|_1 \qquad (12)$$

### A.3  TABLE OF LABEL EFFICIENCY

Table 4: Label efficiency in terms of AP3D.

| Methods | Backbone | Pre-training | Tuned Module. | 100% data | 50% data | 10% data |
|---|---|---|---|---|---|---|
| CubeRCNN | DLA34 | ImageNet cls. | DLA34+3D Head | 31.75 | 25.32 | 7.83 |
| DreamTchr | ResNet50 | SD distill | Res50+3D Head | 33.20 | 26.61 | 8.45 |
| DIFT-SD | StableDiff | LION5B gen. | 3D Head | 28.86 | 24.94 | 7.91 |
| 3DiffTection | StableDiff+Geo-Ctr | Aktsn nvs. | 3D Head | 30.16 | 27.36 | 14.77 |
| 3DiffTection | StableDiff+Geo-Ctr | Aktsn nvs. | Sem-Ctr+3D Head | 39.22 | 35.48 | 17.11 |

The label efficiency table is shown in Tab. 4. Please refer to the Experiment section of the main text for the analysis of label efficiency experiments.

---

[1] https://github.com/huggingface/diffusers/blob/main/examples/controlnet/train_controlnet.py

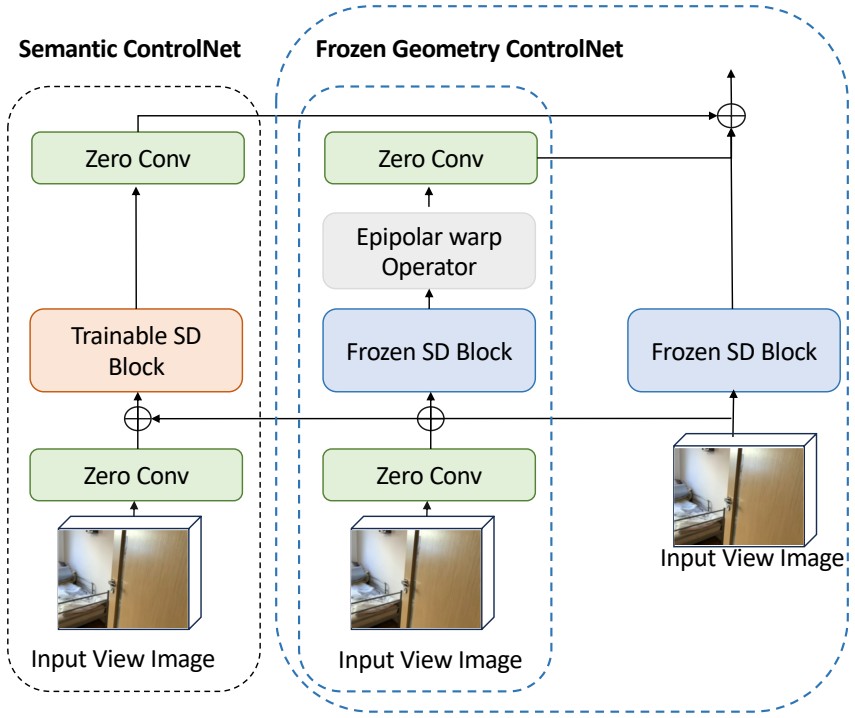

Figure 6: **Semantic ControlNet.** When tuning Semantic ControlNet, we freeze the pre-trained Geometric ControlNet and the original Stable Diffusion block. For both training and inference, we input the identity pose into the Geometric ControlNet by default.

### A.4 FIGURE OF SEMANTIC CONTROLNET

The figure of Semantic ControlNet is depicted in Fig. 6. Please refer to the main description of Semantic ControlNet in the method part.

### A.5 MORE VISUALIZATION

**Visualization of 3D bounding box.** We show more visualization results in this section. Specifically, Fig. 7 and Fig. 8 are the 3D bounding box visualisation results on the test sets of Omni3D-ARKitScenes and Omni3D-SUN-RGBD, respectively. We observe that our 3DiffTection can predict more accurate 3D bounding boxes compared to Cube-RCNN. More importantly, 3DiffTection does not fail even in very challenging cases, such as the second column of Fig. 7 and the first column of 8. They show that 3DiffTection is able to handle occlusions where the chairs are occluded by the tables.

**Visualization of novel-view synthesis.** We then provide more visualization results about novel-view synthesis, as shown in Fig. 10. We randomly choose the images that are never seen during the training of geometric ControlNet. To provide how the camera rotates, we present the warped images as well. Even though novel-view synthesis at the scene level is not our target task, it can be seen that our model can still generalize well under the challenging setting of synthesising novel view from one single image.

**Visualization of 3D correspondences.** We provide more visualization results about 3D correspondences, as shown in Fig. 9.

### A.6 LATENCY

We evaluate the latency of our proposed method on one RTX 3090 GPU. The latency comparison is shown in Tab. 5.

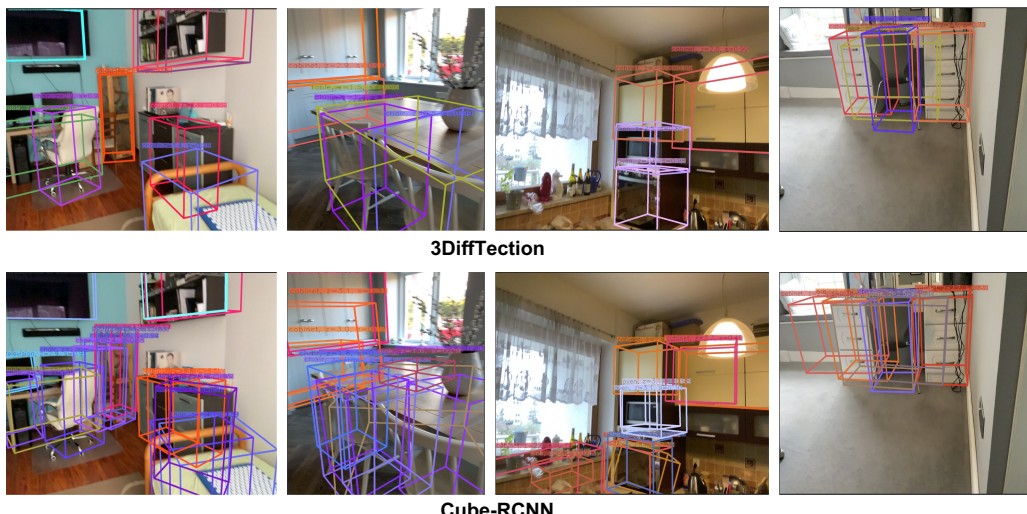

Figure 7: Visualization of 3D bounding boxes on the Omni3D-ARKitScenes test set.

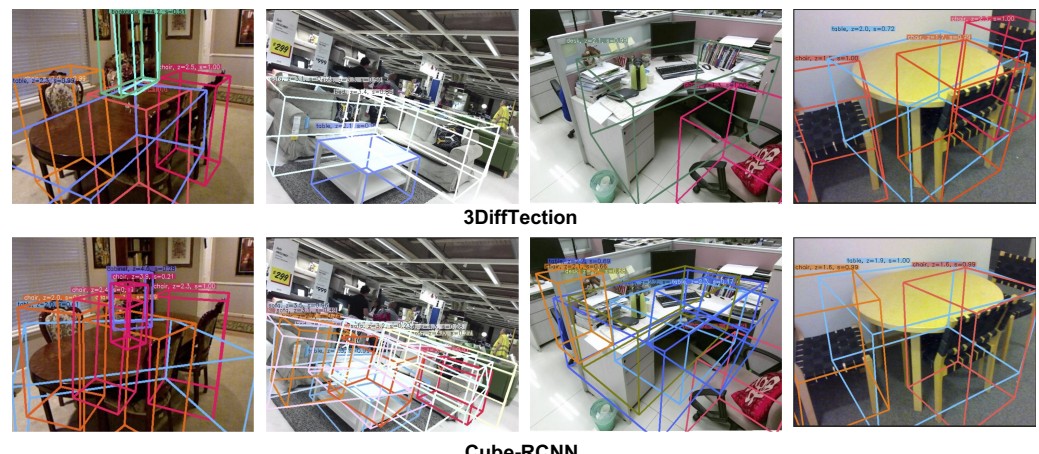

Figure 8: Visualization of 3D bounding boxes on the Omni3D-SUNRGB-D test set.

Table 5: Latency comparison on one 3090Ti GPU.

| Method | Latency (s) |
|---|---|
| 3DiffTection (w/o SemanticControlNet) | 0.104 |
| 3DiffTection | 0.133 |
| 3DiffTection (w/ 6 virtual view Ensemble) | 0.401 |
| Cube-RCNN-DLA34 | 0.018 |

### A.7 DETECTION RESULTS ON SUN-RGBD COMMON CLASSES

We also evaluate our method on the common SUN-RGBD 10 classes, as shown in Tab. 6, as following (Brazil et al., 2023). Experiments demonstrate that our proposed 3DiffTection significantly improves the previous method by a large margin.

### A.8 CROSS-DATASET EVALUATION ON SUN-RGBD DATASET

### A.9 CORRESPONDENCE EVALUATION BASED ON SUPERGLUE.

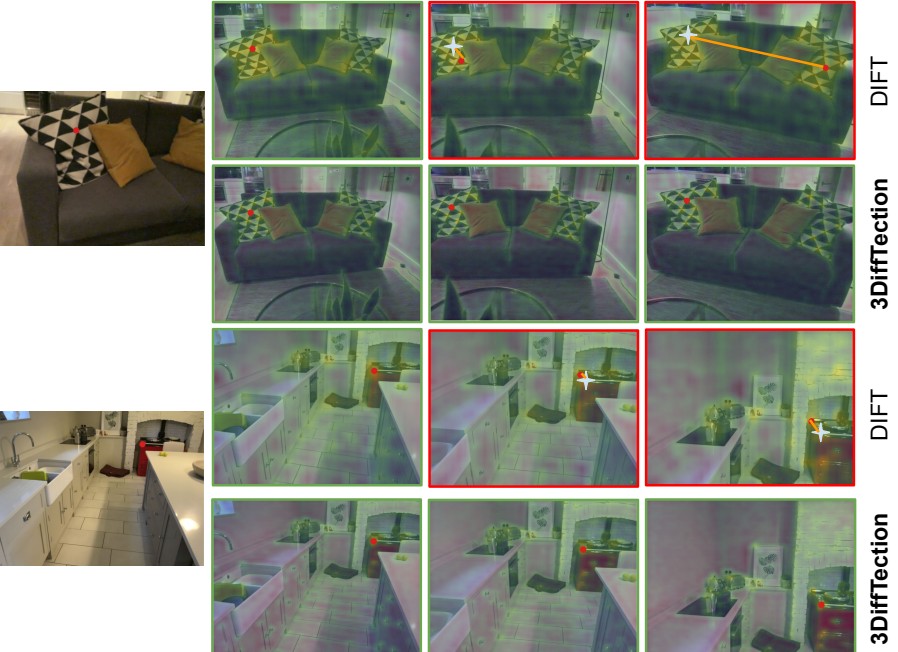

Figure 9: **Visualization of 3D correspondences prediction using different features**. Given a **Red Source Point** in the leftmost reference image, we predict the corresponding points in the images from different camera views on the right (**Red Dot**). The ground truth points are marked by **Blue Stars**. Our method, 3DiffTection, is able to identify precise correspondences in challenging scenes with repetitive visual patterns. The orange line measures the error of the prediction and ground truth points.

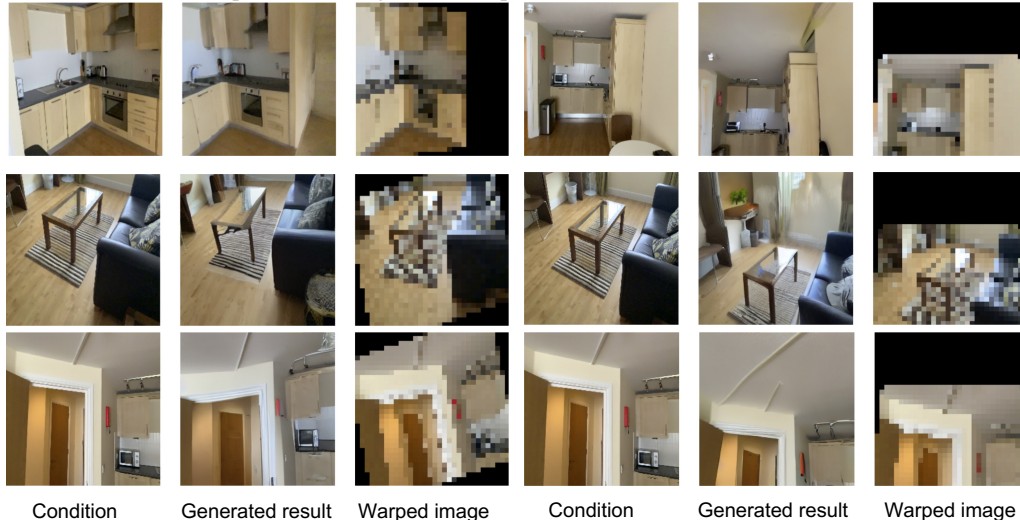

| Condition | Generated result | Warped image | Condition | Generated result | Warped image |

Figure 10: Visualization of novel-view synthesis. We rotate the camera by 15 deg anchoring to different axises. The warp image can be used to indicate the camera rotated directions.

Table 6: Comparison on common categories of SUN-RGBD dataset.

| Method | AP3D |
|---|---|
| Total3D (Nie et al., 2020) | 27.7 |
| ImVoxelNet (Rukhovich et al., 2022) | 30.6 |
| Cube-RCNN | 35.4 |
| 3DiffTection | 38.8 |

| Methods | Backbone | Pretrained on ARKit | Tuned on SUNRGBD | Zero-shot(w/o 2D GT) | Zero-shot(w/ 2D GT) |
|---|---|---|---|---|---|
| DIFT-SD | StableDiff | ✗ | 21.92 | 16.74 | 25.31 |
| CubeRCNN | DLA34 | ✓ | 22.72 | 16.81 | 25.05 |
| 3DiffTection | StableDiff+Geo-Ctr | ✓ | 23.11 | 17.37 | 26.94 |
| 3DiffTection | StableDiff+Geo-Ctr+Sem-Ctr | ✓ | 27.81 | 22.64 | 30.14 |

Table 7: **Cross-domain experiment on the Omni3D-SUNRGBD dataset.** The "Pre-trained on ARKit" denotes we pre-train the **backbone** on Omni3D-ARKitscenes. For CubeCNN, we pre-train it with 3D detection supervision. For all zero-shot experiments, the methods are first trained on Omni3D-ARKitscenes for 3D detection and then directly tested on Omni3D-SUNRGBD dataset. "2D GT" means we use ground-truth 2D bounding box to crop ROI image features. The results are reported for overlapped 14 classes between Omni3D-SUNRGBD and Omni3D-ARKiSscenes dataset.

| Methods | AUC@5 | AUC@10 | AUC@20 | Precision | MS |
|---|---|---|---|---|---|
| DIFT-SD | 5.61 | 6.15 | 14.73 | 16.95 | 4.57 |
| 3DiffTection | 7.49 | 8.92 | 18.66 | 20.11 | 6.93 |
| SuperPoint+SuperGlue | 14.77 | 38.45 | 55.99 | 72.84 | 19.26 |
| SuperPoint+3DiffTection+SuperGlue | 25.86 | 45.13 | 61.41 | 72.11 | 19.13 |

Table 8: Correspondence evaluation on ScanNet dataset. We use the sampled data provided by SuperGlue and use Superpoint to find the key-points for evaluation. The first group represents the fair comparison between DIFT and our 3DiffTection. The second group represents the results the we fuse 3DiffTection features and SuperPoint features together.

