# OpenReview forum: "3DiffTection: 3D Object Detection with Geometry-Aware Diffusion Features"
_ICLR.cc/2024/Conference — ICLR 2024 Conference Withdrawn Submission_

### Official Review · Reviewer_CDfS · 2023-10-26

**Soundness:** 3 good
**Presentation:** 4 excellent
**Contribution:** 3 good
**Rating:** 6
**Confidence:** 4

**Summary:**

This paper introduce 3DiffTectio, a 3D detection model using posed images, based on a generative diffusion model. It overcomes the limitations of current diffusion models in 3D tasks and leverage two controlnet to refine the diffusion feature to be 3D-aware, ultimately excelling at identifying cross-view correspondences. The proposed method outperforms predecessors like Cube-RCNN by 9.43% on a specific dataset and showcases impressive data efficiency and cross-domain generalization.

**Strengths:**

1. This paper is quite novel, revealing that the features of generative models are also suitable for downstream perception tasks.
2. The figures and datasets chosen in the paper effectively elucidate its motivation and the viability of the proposed method.
3. The performance is quite good.

**Weaknesses:**

1. I am quite doubt whether the geometric ControlNet truly introduces 3D awareness. Although they trained the ControlNet on posed images using novel view synthesis, the inclusion of a warping operation in the ControlNet suggests that the diffusion model is simply performing an image completion on the warped features.
2. The method is trained on video data, which means it posses the piror knowledge on general 3D scene. In contrast, the baseline method has not been trained on posed images, making this comparison somewhat unfair.
3. For perception tasks, the size of the model and its runtime need to be considered. The combination of ControlNet + diffusion might make the model inefficient.

**Questions:**

See weakness.

---

> ### Author Response · Authors · 2023-11-20
> **Response**
>
> We thank you for your comments and appreciation to our paper. We respond to your concern below.
>
> **To respond the baseline method has not been trained on posed images, while we train the model on posed images:**
> We refer the reviewer to see our table. 1.
> We compare our method with NeRF-Det and ImvoxelNet, which are both trained on the posed images. Actually, these methods use stronger priors with more posed images while we only use two posed images during training, our 3DiffTection can still outperform them on single-image 3D object detection.
>
>
> **To respond to whether geometric ControlNet truly introduces 3D awareness:**
> Please refer to our correspondence visualization results, and we provide more visualization in our appendix as well as our quantitative correspondence evaluation. Note that the correspondence evaluation all come from geometric controlnet with input identical pose, and it never accesses correspondences ground truth, which means we have a totally fair comparison with the original DIFT model. **With this context, our method can find correspondences between multiview images without input pose, while the DIFT model fails. If our model does not learn 3D awareness, this means it can not find the accurate correspondences, and during novel-view synthesis, it can not accurately warp the features to the correct places.** The current correspondence experiments and novel-view synthesis experiments demonstrate the effectiveness of our model learning 3D awareness.
>
> **To respond the efficiency**
> The efficiency is currently out of scope but it is an important topic. Currently, there are some efforts on this track, for example, Q-Diffusion [1], especially being efficient during inference. We will also put some efforts on this track in the future.
>
> [1]Q-Diffusion: Quantizing Diffusion Models. ICCV2023

---

### Official Review · Reviewer_T1bR · 2023-11-01

**Soundness:** 2 fair
**Presentation:** 2 fair
**Contribution:** 2 fair
**Rating:** 5
**Confidence:** 4

**Summary:**

The paper introduces "3DiffTection," an advanced methodology for 3D detection from posed images, leveraging features from a 3D-aware diffusion model. The approach adeptly addresses the challenges associated with annotating large-scale image data for 3D object detection. By integrating geometric and semantic tuning strategies, the authors have augmented the capabilities of existing diffusion models, ensuring their applicability to 3D tasks. The method notably surpasses previous benchmarks, demonstrating high label efficiency and robust adaptability to cross-domain data.

**Strengths:**

1. The methodology effectively circumvents the challenges of annotating large-scale image data for 3D object detection.
2. Through the integration of geometric and semantic tuning strategies, the authors have enhanced the capabilities of diffusion models

**Weaknesses:**

1.The performance on a broader range of datasets is missing, and it should also be compared with more recent research.

2.Semantic ControlNet lacks a more comprehensive analysis.

**Questions:**

1.Could you provide further explanations regarding how Semantic ControlNet and Novel View Synthesis assist in enhancing models, along with corresponding analyses?

2.Could you present comparative performance results of the more models across the more datasets?

1. In the stage2, is the input noised source image or pure gaussian noise?
2. is there any other generative model can get the same improvement by embedding geometry and semantic control?

---

> ### Author Response · Authors · 2023-11-20
> **Explanation how novel view synthesis and semantic controlNet help 3d detection**
>
> We sincerely appreciate your time and efforts form reviewing our paper.
>
> To respond to how semantic controlnet and novel view synthesis assist in enhancing the models:
>
> **Why does novel view synthesis help enhance the 3D object detection model?**
> Novel-view synthesis is a task that “warps” the pixels of the condition image into the target one, which inherently enforces the model to learn the 3D structures of the scenes, otherwise the shifting would be wrong. Learning 3D information is important for 3D object detection. The similar spirit is also proposed in NeRF-Det. To demonstrate that our geometric ControlNet instills 3D awareness in the model, we compare the SD features with and without our geometric ControlNet. For the comparison, we use ScanNet, which allows us to measure performance against ground truth and report accuracy. Moreover, it is important to note that ScanNet was not used for training at any point, ensuring a fair comparison. The correspondence analysis can demonstrate the hypothesis above. If the model learns the pixel shifting, it should have been expressed by correspondences. To be more specific, we conduct quantitative correspondence experiments, as shown in the following table. Experiments also demonstrate that training a geometric controlnet with novel-view synthesis task can improve the correspondence results.
>
> Correspondence experiment on ScanNet dataset.
>
>
> |Methods     |   AUC@5    |   AUC@10      |     AUC@20    |  Precision   |  MS |
> -------------- -------------- ----------------- ---------------- -------------- -------
> |DIFT-SD      |     5.61       |        6.15         |     14.73         |  16.95         | 4.57|
> -------------- -------------- ----------------- ---------------- -------------- -------
> |3DiffTection|      7.49       |        8.92         |     18.66       |     20.11        | 6.93|
>
>
> **Why does Semantic-ControlNet help enhance the 3D object detection model?**
> The motivation for utilising another semantic controlnet is that the features from stable diffusion + geometric controlnet are designed for image generation and aren’t specifically fine-tuned for 3D detection. Therefore, there is further potential for optimization towards detection. We want to emphasize that tuning the controlnet is also a way of parameter-efficient finetuning that does not break the original representations from pretrained Stable Diffusion model parameters. Maintaining parameter efficiency is crucial for achieving label efficiency, a key objective in our research, as demonstrated in our label-efficiency experiment.

---

> > ### Author Response · Authors · 2023-11-20
> > **Response to more experiments and remaining questions**
> >
> > **Q: In the stage2, is the input noised source image or pure Gaussian noise?*(
> > A: A noised image. Note that our task is to detect objects in images, during detection, a single denoising pass is performed for feature extraction, and the detection head proceeds to predict the 3D box. generation from pure gaussian noise is not required.
> >
> > **Q: is there any other generative model can get the same improvement by embedding geometry and semantic control?**
> > A: We mainly build our algorithm on top of stable diffusion because it is the state-of-the-art method which demonstrated having useful features for perception tasks, and because ControlNet equipts is with a parameter efficient conditioning ability.  We cannot think of another model with similar properties and would be glad if the reviewer could provide us with more suggestions of what kinds of different generative models.
> >
> > **More experiments:** We conduct more experiments about showing the zero-shot ability on SUNRGBD dataset. The performance is shown in the following table.
> >
> > Methods       | Backbone   | Pretrained on ARKit |Tuned on SUNRGBD | Zero-shot(w/o 2D GT)| Zero-shot(w/ 2D GT)|
> > --------------- -------------- -----------------------   ------------------------ -------------------------- ------------------------
> > DIFT-SD        |  StableDiff  |                ✗             |     21.92                       |                 16.74           |          25.31|
> > --------------- -------------- -----------------------   ------------------------ -------------------------- ------------------------
> > CubeRCNN  |  DLA34       |               ✓              |      22.72                     |                  16.81          |          25.05|
> > --------------- -------------- -----------------------   ------------------------ -------------------------- ------------------------
> > 3DiffTection  |StableDiff+Geo-Ctr|  ✓                 |      23.11                      |                  17.37          |          26.94|
> > --------------- -------------- -----------------------   ------------------------ -------------------------- ------------------------
> > 3DiffTection |StableDiff+Geo-Ctr+Sem-Ctr| ✓   |      27.81                      |                   22.64         |          30.14|
> >
> >
> > To assess the capability of geometric ControlNet to carry its 3D awareness to other datasets, we employed a 3DiffTection model with its geometric ControlNet trained on the OMni3D-ARKitscene dataset, and conduct cross-dataset experiments on the Ommni3D-SUNRGBD dataset. We evaluate it with two settings: (1) finetune the parameters on the Omni3D-SUNRBGD dataset and test the performance on Omni3D-SUNRGBD dataset (the fourth column), and (2) train the parameters on the Omni3D-ARKitscenes dataset and directly test the performance on Omni3D-SUNRGBD dataset in a zero-shot setting. (The fifth and the sixth column). **Experiments demonstrate the superior transferability of our 3Difftection.**

---

### Official Review · Reviewer_16V1 · 2023-11-08

**Soundness:** 3 good
**Presentation:** 2 fair
**Contribution:** 2 fair
**Rating:** 3
**Confidence:** 4

**Summary:**

The manuscript addresses the task of 3D object detection from posed images by leveraging the 2D feature space of pre-trained large diffusion models and exploiting ControlNet to integrate 3D geometric awareness and auxiliary semantic infomation. Extensive experiments on Omni3D datasets demonstrate the effectiveness of the method.

**Strengths:**

* The manuscripts first proposes to improve 3D awareness by aggregating features with ControlNet from auxiliary views
* The method proposed in the manuscript achieve significant margins over comparable baselines.

**Weaknesses:**

* The novelty seems limited. Though with the insight of integrating 3D awareness and closing the domain gap with auxiliary semantic information, the actual practice is adopting existing work ControlNet (Zhang et al., 2023)[^1]. The proposed method is more like an application of ControlNet on a specific task (in this case, the task of 3D object detection from posed images).
* The sampling strategy on the epipolar line needs clarification. If the line of sight is blocked by objects, it is unreasonable to include features sampled behind the blocking objects. It is recommended to provide more details on how to avoid aggregate sampling features from blocked views.
* Minor problems in presentation. *Diffusiondet: Diffusion model for object detection* appears twice in the *Reference* section

[^1]: Lvmin Zhang, Anyi Rao, and Maneesh Agrawala. Adding conditional control to text-to-image diffusion models, 2023.

**Questions:**

See *Weaknesses* section.

---

> ### Author Response · Authors · 2023-11-15
> **Response to Novelty**
>
> We thank the reviewer for the time in reading our paper.
>
> We respectfully disagree with the stated main weakness, i.e, that the proposed method is merely an application of ControlNet. On an intellectual level, a successful utilization of a technique, unless trivial, can still be interesting to the audience for advancing science. For the case of our specific work, our approach introduces ideas from ControlNet in an advanced way and in a completely different manner than originally proposed (3D detection rather than conditional generation). We cannot think of a way to trivially use Controlnet to achieve the same task as indicated by the reviewer, and would appreciate a more concrete proposal that we can discuss.
>
> We provide further explanation below and invite the reviewer to take another careful read of our paper:
> 1. Our work aims at 3D object detection. Our method highlights that (1) diffusion features are semantically meaningful for use in perception tasks. However, they are not good enough for 3D tasks out of the box. Therefore, (2) we leverage ControlNet to instill 3D awareness into the learned representation. With these improved features we proceed to (3) tune them along with a 3D detection head.
>
> 2. As discussed in Sec 3.2, we find that ControlNet’s design is inherently biased towards condition signals that are pixel-aligned with the generated image. Indeed, our early experiments indicate that directly conditioning on the source image fails at generating useful novel views. To overcome this, we propose to modify ControlNet with an epipolar warp operator. This simple operator makes a big difference. Experimentally, please refer to our Tab. 2 (Row 5), where we remove the warp operator and reduce geometric ControlNet to the original ControlNet. The results demonstrate that a simple use of ControlNet does not work for 3D object detection.
>
> 3. Finetuning Semantic ControlNet for specific tasks and data itself is also novel. As far as we know, previous works only use ControlNet for generation tasks, we tune it for perception tasks, which is also a new use.
>
> 4. Finally, our method leverages both geometric and semantic tuning stages to enable virtual view augmentation, for enhanced detection prediction.
>
> We hope this explanation makes it clear that equating our method to a mere application of ControlNet is a drastic oversimplification. Our approach involves a plethora of insights into the premise and limitations of diffusion models for 3D perception tasks, an understanding of the strengths and weaknesses of ControlNet, geometric reasoning, and drastic improvements in single-image 3D detection over past work. Thus, we believe that beyond the compelling results, readers will also acquire new insights and ideas for future work.

---

> > ### Author Response · Authors · 2023-11-15
> > **Sampling method**
> >
> > We shoot a ray through each target view pixel, uniformly sample points along that ray and project the points to the source view. Our aggregator aims to utilize the source view features to provide useful conditioning cues for generating the target image. These features are derived from a diffusion model at varying levels of coarseness. We believe they contain enough information to enable view synthesis that accounts for occlusions. Interestingly, in our study, we experimented with a more sophisticated aggregator than simple averaging by employing a transformer. However, we did not observe a noticeable improvement in generation quality.
> >
> >
> > We are preparing the revision of the submission and will update the reviewers with more visualization and experiment results, including qualitative and quantitative correspondence checking etc